# I329L: A Dual Action Viral Antagonist of TLR Activation Encoded by the African Swine Fever Virus (ASFV)

**DOI:** 10.3390/v15020445

**Published:** 2023-02-05

**Authors:** Sílvia Correia, Pedro Luís Moura, Sónia Ventura, Alexandre Leitão, Robert Michael Evans Parkhouse

**Affiliations:** 1Instituto Gulbenkian de Ciência, 2780-156 Oeiras, Portugal; 2CIISA—Centro de Investigação Interdisciplinar em Sanidade Animal, Faculdade de Medicina Veterinária, Universidade de Lisboa, 1300-477 Lisbon, Portugal; 3Laboratório Associado para Ciência Animal e Veterinária (AL4AnimalS), 1300-477 Lisbon, Portugal

**Keywords:** African swine fever virus, I329L protein, toll-like receptor, immune evasion, innate immunity

## Abstract

The African Swine Fever Virus (ASFV) is an economically important, large DNA virus which causes a highly contagious and frequently fatal disease in domestic pigs. Due to the acute nature of the infection and the complexity of the protective porcine anti-ASFV response, there is no accepted vaccine in use. As resistance to ASFV is known to correlate with a robust IFN response, the virus is predicted to have evolved strategies to inhibit innate immunity by modulating the IFN response. The deletion of virus host evasion gene(s) inhibiting IFN is a logical solution to develop an attenuated virus vaccine. One such candidate, the ASFV ORF I329L gene, is highly conserved in pathogenic and non-pathogenic virus isolates and in this study we confirm and extend the conclusion that it has evolved for the inhibition of innate immunity initiated through Toll-like receptors (TLRs). Specifically, the ASFV I329L extracellular (ECD) and intracellular (ICD) domains inhibit TLR signalling by two entirely different mechanisms. Bioinformatics modelling suggests that the ECD inhibits several TLR signalling pathways through a short sequence homologous to the conserved TLR dimerization domain, here termed the putative dimerization domain (PDD). Remarkably, both full length and PDD constructs of I329L were demonstrated to inhibit activation, not only of TLR3, but also TLR4, TLR5, TLR8 and TLR9. Additionally, the demonstration of a weak association of I329L with TLR3 is consistent with the formation of a non-signalling I329L-TLR3 heterodimer, perhaps mediated through the PDD of I329L. Finally, the ICD associates with TRIF, thereby impacting on both TLR3 and TLR4 signalling. Thus, I329L offers potential as a general inhibitor of TLR responses and is a rational candidate for construction and testing of an I329L deletion mutant vaccine.

## 1. Introduction

The African Swine Fever Virus (ASFV) is a large, cytoplasmically replicating DNA virus which causes a highly contagious and frequently fatal acute disease in domestic pigs [1]. This disease, endemic in many African countries, entered Portugal and Spain in the past with disastrous economic consequences. More recently, in 2007, the virus entered Georgia, in the Caucasus, and has now spread essentially worldwide [2]. There is no routinely used vaccine for ASFV and control relies on restricting animal movement and slaughter of infected pigs. This is particularly difficult in areas where both the wildlife reservoirs (warthog, bushpig and wild boar) and the tick vector of the virus are present, since these animal reservoirs may be persistently infected for many years. Development of an efficient and safe vaccine against ASFV would protect Europe from accidental entry of the virus, and remove a major pathogen of pigs from large endemic areas of Africa, where the increasing frequency of outbreaks has caused devastating losses to both rural and commercial farms.

Due to the acute nature of the infection and the complexity of the protective porcine anti-ASFV immune response [3], the development of vaccine candidates has evolved slowly [4]. So far, inactivated and subunit vaccines have shown poor results and a live, replicating virus vaccine attenuated by the deletion of virus host evasion gene(s) presents a practical solution with the advantage of stimulating both, innate and adaptive immunity.

As resistance to ASFV is known to correlate with a robust IFN response [5], the virus is predicted to have evolved a number of counter-strategies to inhibit the IFN response, a prediction that has been verified [6]. Therefore, deletion of virus genes inhibiting induction and/or impact of IFN is a logical choice for the construction of such an attenuated gene deletion virus vaccine. In our previous work we have shown that the protein encoded by the ASFV ORF I329L impairs the porcine, murine, and human TLR3-mediated cellular responses that lead to activation of both IFN-β promoter and NF-κB and thus induction of type I IFN and inflammatory cytokines [7].

The ASFV I329L gene codes for a non-essential [8] Type I membrane protein without any immediately obvious sequence homology, and which is highly conserved in both pathogenic and non-pathogenic virus strains. Functional assays, however, demonstrated that I329L not only inhibited the activation of the IFN-β promoter and NF-κB, but also inhibited the secretion of IFN [3]. More sophisticated bioinformatics structural modelling now suggests that (1) The extracellular domain (ECD) of I329L might serve as a dsRNA decoy or hinder TLR3 signalling, perhaps through formation of an I329L-TLR3 heterodimer, and (2) The intracellular domain (ICD) might interact with the downstream intracellular signalling intermediate TRIF [9].

In this work, using a combination of biochemical and bioinformatics techniques, we confirm and extend the conclusion that the ASFV ORF I329L has evolved for the inhibition of innate immunity initiated through TLRs through two entirely different mechanisms. Protein structure simulation suggests that the ECD interferes with TLR signalling through a short sequence corresponding to the conserved TLR dimerization domain and termed the putative dimerization domain (PDD). Remarkably, both the full length and PDD constructs of I329L were demonstrated to inhibit activation not only of TLR3, but also TLR4, TLR5, TLR8 and TLR9. In addition to the inhibitory impact of the ECD, the ICD associates with the intracellular signalling intermediate TRIF, thereby impacting on both TLR3 and TLR4 signalling. Finally, bioinformatic analysis also suggested that both the ECD and ICD could inhibit TLR3 signalling. Thus, I329L offers potential as (1) An inhibitor of TLR responses and (2) A rational candidate for construction and testing of an I329L deletion mutant vaccine.

## 2. Materials and Methods

### 2.1. Homology Modelling of I329L

The sequences of I329L and its ECDTM and ICD domains were taken from the UniProt database (accession number A9JLD8) and investigated for possible homologies using Phyre2 (Protein Homology/analogY Recognition Engine V2.0 Webserver) [10] to predict homologies with proteins of known structure. A preliminary set of models was obtained by use of the I-TASSER protein prediction server [11] and used as a template for more accurate modelling. The resulting I329L structures were homology-modelled with Modeller 9.14 [12] (http://zhang.bioinformatics.ku.edu/I-TASSER, accessed on 31 December 2017) using the pl329L alignment with PDB templates obtained from Phyre2 and I-TASSER. The quality of the resulting models was assessed using PROCHECK v3.5.4 [13], and the best quality models were used to determine possible protein complexes by superimposition with the structure of TLR3. The membrane was modelled using palmitoyl-oleoyl-phosphatidylethanolamine (POPE) molecules. Ray-traced images were obtained with PyMOL v1.7.2 RC2, https://pymol.org/2/, version 1.7.2, accessed on 31 December 2017.

### 2.2. Cell Culture

African green monkey Vero and HEK-293T cells were cultured in Dulbecco’s Modified Eagle’s Medium (DMEM) (Gibco, Carlsbad, CA, USA) supplemented with 100 U/mL penicillin G sodium/100 μg/mL streptomycin sulfate (Gibco) and 10% (*v*/*v*) heat-inactivated fetal bovine serum (FBS) (Gibco).

### 2.3. Plasmids

The I329L ectodomain with the transmembrane region (ECDTM) (aminoacids 1 to 260), the putative dimerization domain with the transmembrane region (PDDTM) (aminoacids 149 to 260), the intracellular domain (ICD) (aminoacids 261 to 329), were amplified by PCR from the pcDNA3-I329L-HA construct [7] and cloned into the pcDNA3-HA plasmid. All constructs contained a C-terminal HA “immunotag” sequence to allow confirmation of protein expression.

The human cytomegalovirus (CMV)–vesicular stomatitis virus G envelope protein, the packaging plasmid pCMVR8.9 and the vector pHR-CMV-eGFP constructs have been described previously [14]. For construction of a recombinant lentivirus vector (pHR-CMV-I329LeGFP), I329L was excised from pCDNA3, together with the HA tag and cloned into the vector pHR-CMV-eGFP upstream of an internal ribosome entry site-driven enhanced green fluorescent protein gene (eGFP).

The luciferase reporter plasmid containing the sequence of the IFN-β promoter [pIFΔ(−125/+72)lucter], [p(PRDII)5tkA(-39)lucter] were gifts of Dr. S. Goodbourn (St George’s Biomedical Sciences Research Centre, University of London, UK). The pCMVβ plasmid contains a β-galactosidase gene that serves as an internal control for transfection efficiency.

The pCMV-1Flag fused to TRIF, hTLR3, hTLR5, hTLR8 were gifts from Dr A. Bowie (Trinity Biomedical Sciences Institute, Trinity College Dublin, Ireland).

The EEA1 Ct-tomato plasmid was a gift from Professor H. Stenmark (Department of Biochemistry, The Norwegian Radium Hospital, Norway).

### 2.4. Production and Measurement of Lentivirus

Lentivirus was produced by transient transfection of HEK-293T cells with a weight ratio of 3:1:1 of vector to packaging to envelope plasmids using Fugene 6 (Roche, Basel, Switzerland) according to the manufacturer’s instructions. Control lentivirus was produced by co-transfection of the packaging and envelope plasmid together with the empty pHR-CMV-eGFP plasmid. For production of recombinant lentivirus expressing I329L, the plasmid pHR-CMV-I329L- eGFP was used. Supernatants containing the lentivirus were collected at 48, 72 and 96 h post-transfection and clarified by low-speed centrifugation, and the lentivirus was collected by ultracentrifugation (25,000 r.p.m. in an SW28 rotor in a Beckman centrifuge). Virus pellets were resuspended in fresh culture medium and frozen at −80 °C. Lentivirus titres were measured by infection of HEK-293T cells with a dilution factor of 4. Analysis of lentivirus-infected cells was performed by detecting eGFP-positive cells by flow cytometry at 48 h post-infection (p.i.).

### 2.5. Immunofluorescence

Vero cells grown on coverslips were transfected using X-treme9 (Roche) with pcDNA3-HA empty vector or pcDNA3-I329L-HA and EEA1 Ct-tomato plasmid (to visualize early endosome marker). At 48 h post-transfection, the cells were fixed with 4% paraformaldehyde in PBS for 15 min at room temperature. The cells were permeabilized with PBS containing 0.1% Triton X-100 for 20 min, washed and blocked with PBS containing 5% normal goat serum and 0.05% Tween 20 for 30 min. In an alternative protocol, the fixation and permeabilization step were omitted in order to stain cell surface proteins. The cells were then incubated with a rat monoclonal anti-HA (SIGMA, St. Louis, MO, USA), and then anti-mouse cy5 (Jackson, Chicago, IL, USA) to visualize the HA tagged I329L, and the cell nuclei were stained with DAPI.

### 2.6. Luciferase Reporter Gene Assay

To define the impact of I329L domains in type I IFN induction via TLR3/TRIF, HEK-293T cells were co-transfected with the IFN-β luciferase reporter plasmid, a TLR3 expression plasmid, the β-galactosidase internal control plasmid and either the empty vector pcDNA3HA (EV), pcDNA3-I329L-HA, pcDNA3-ECDTM-HA, pcDNA3-ICD-HA or pcDNA3-PDDTM-HA plasmids, as indicated, according to the Lipofectamine 2000 (Invitrogen) protocol. Cells were either induced with Poly(I:C) (Amersham, Stafford, UK) (25 μg/mL), ectopic TRIF (pCMV-1Flag-TRIF) expression or left untreated (medium).

To investigate the impact of I329L and its domains on signalling through TLR5 and TLR8, HEK-293T cells were co-transfected with the NF-κB (PRDII element)-luciferase reporter plasmid, the β-galactosidase internal control plasmid, and either TLR5 or TLR8 plasmids, together with either empty vector pcDNA3HA (EV), pcDNA3-I329L-HA, pcDNA3-ECDTM-HA or pcDNA3-PDDTM-HA plasmids, as indicated, according to the Lipofectamine 2000 (Invitrogen, Waltham, MA, USA) protocol. Cells were either induced with 100 ng/mL Flagellin (Alexis Biochemicals, Lausen, Switzerland) for the TLR5 transfected cells or 1 μg/mL R848 (Invivogen) for the TLR8 transfected cells. Control cells were left untreated.

In an alternative protocol, to test the impact of I329L and its domains on TLR4 and TLR9 signalling pathways, Vero cells were co-transfected with the NF-κB (PRDII element)-luciferase reporter plasmid, the β-galactosidase internal control plasmid, and either the empty vector pcDNA3HA (EV), pcDNA3-I329L-HA, pcDNA3-ECDTM-HA or pcDNA3-PDDTM-HA plasmids, as indicated, according to the Lipofectamine 2000 (Invitrogen) protocol. Cells were either induced with 100 ng/mL LPS (Invivogen) to stimulate TLR4, or 10 μg/mL ODN 2006 (Class B CpG) (Invivogen) to stimulate TLR9. Control cells were left untreated.

Luciferase activity was normalized to β-galactosidase activity as a control for transfection efficiency. Data are expressed as means of relative luciferase units (RLU) ± SD of triplicate culture wells, and are representative of three independent experiments. Statistically significant difference was assessed by Student’s *t*-test and is represented as *p* ≤ 0.05 (*).

### 2.7. Immunoprecipitation to Demonstrate Association of I329L and TLR3

HEK-293T cells were stably infected with HA-I329L recombinant lentivirus (pHR-CMV-I329LeGFP), and as a negative control, with the control “empty” recombinant lentivirus (pHR-CMV-eGFP), and then transfected with TLR3-Flag expression plasmid, according to the Fugene 6 (Promega, Madison, WI, USA) protocol. At 48 h post-transfection, the cells were harvested and lysed in 1% digitonin lysis buffer (1% digitonin, 50 mM Iodoacetamide and 20 mM Tris-HCl (pH 8.0)) containing a protease inhibitor cocktail (SIGMA). Immunoprecipitations were performed with Dynabeads protein G (Millipore, Burlington, MA, USA) and 0.5% digitonin washing buffer (0.5% digitonin and 35 mM Tris-HCl (pH 8.0)) containing a protease inhibitor cocktail (SIGMA), using mouse anti-human TLR3 (Thermofisher, Waltham, MA, USA) and mouse anti-HA (SIGMA). Immunoprecipitates were resolved on a 10% SDS-PAGE gel and Western blot was performed using: mouse anti-human TLR3 (Thermofisher) followed by horseradish peroxidase goat anti-mouse secondary antibodies (Invitrogen), or rat anti-HA-HRP conjugated (Roche) to detect I329L.

### 2.8. Immunoprecipitation to Demonstrate Association of ICD-I329L and TRIF

HEK-293T cells were transfected with pcDNA3-HA (EV) or pcDNA3-ICD-HA (ICD), according to the Fugene 6 (Promega) protocol. 48 h post-transfection, the cells were harvested and lysed in 1× lysis buffer (Cell Signaling, Danvers, MA, USA) containing Triton X-100 and a protease inhibitor cocktail (SIGMA). Immunoprecipitations were performed with Dynabeads protein G (Millipore) using rabbit anti-human TRIF antibody (Cell Signaling), and rabbit anti-HA (SIGMA). Immunoprecipitates were resolved on a 12% SDS-PAGE gel and detection on Western blot was performed using: rabbit anti-human TRIF (Cell Signaling) followed by horseradish peroxidase goat anti-rabbit secondary antibodies (Invitrogen), or rat anti-HA-HRP conjugated (Roche) to detect the ICD of I329L.

### 2.9. ELISA

HEK-293T cells were co-transfected with empty vector pcDNA3HA (EV), pcDNA3-I329L-HA, pcDNA3-ECDTM-HA or pcDNA3-PDDTM-HA and TLR3 plasmid, according to Lipofectamine 2000 (Invitrogen) protocol. At 48 h post-transfection the cells were either stimulated with 25 μg/mL of Poly(I:C) for 5 h, or left untreated. Cell supernatants were collected and ELISA for IFN-β was performed according to the manufacturer’s protocol (Human IFN-β bioluminescent ELISA kit from Invivogen). Data are expressed as means ± SD of triplicate wells and are representative of three independent experiments. Statistically significant difference was assessed by Student’s *t*-test and is represented as *p* ≤ 0.05 (*).

## 3. Results

### 3.1. I329L Localizes to the Cell Membrane and Early Endosomes

Previous work developed in our laboratory identified I329L as a novel inhibitor of TLR3 activation [7,9]. As TLR3 localizes in the endosomes and/or cell surface membranes where it interacts with its ligand, double-stranded viral RNA, the cellular localization of I329L was investigated.

Vero cells were transfected with pcDNA3-HA empty vector (EV) or pcDNA3-I329L-HA and EEA1-tomato plasmid, and at 48 h post-transfection examined by fluorescence microscopy for localization of the I329L protein. Importantly, although I329L is partially observed in the cytoplasm, it demonstrated co-localization in punctate structures corresponding to the early endosome as revealed by the EEA1 marker (Figure 1A). To determine if I329L also localized to the cellular membrane, Vero cells were transfected with EV or I329L and examined at 48 h post-transfection for surface membrane localization by fluorescence staining of non-permeabilized cells. Using this strategy, I329L was also detected, through which we deduce its ability to be mobilized to the cell membrane (Figure 1B). Thus, the I329L virus protein constitutively localizes at the structures most appropriate to its observed function as an inhibitor of the TLR3 activation. Importantly, these observations do not exclude a potential function for I329L in inhibiting other TLR signalling pathways.

### 3.2. Homology Modelling of I329L Reveals Distinct Functional Domains

Using homology modelling tools for protein structure simulation (Material and Methods) [10,11,12,13], and based on structural assessments from previous work [9], a complete I329L model was constructed as a platform for subsequent strategies to determine structure-function correlations.

In our first approach, the PHYRE 2.0 and I-TASSER servers were used to search for plausible templates. A distributed approach was used due to the structural heterogeneity of both I329L domains. The I329L ECD was modelled using the Nogo receptor ECD (PDB ID: 1OZN), the decorin protein core (PDB ID: 1XKU), the Lingo-1 ECD (PDB ID: 2ID5), the TLR3 ECD (PDB ID: 3CIY) and the I-TASSER basic I329L model as templates. The I329L ICD was subsequently modelled using the TIR domain of TLR10 (PDB ID: 2J67) as a single template. Finally, after modelling the ECD and ICD, the TM region was modelled as a single α-helix and the three structures were joined to construct a full model.

After further inspection of the TLR3 ECD chain structure [15], two regions of functional homology shared by I329L and TLR3 were identified. These regions, marked in the models shown in Figure 2A,B, correspond to the region responsible for TLR3 dimerization (putative dimerization domain, PDD) and the central domain responsible for dsRNA interaction (dsRNA interaction domain, DID). Importantly, both domains are conserved among TLR3 homologues in other species and required for its correct function [15,16].

Structural superposition of I329L on the TLR3 model permitted the construction of a putative I329L-TLR3 dimer model, also defining the interaction with dsRNA (Figure 2C,D). In this model, it is clear that (1) The I329L ECD only has one possible location for interaction with dsRNA, and (2) It lacks the second dsRNA-binding region observed in the N-terminus of the TLR3 structure. Additionally, a previous bioinformatics study [7,9] observed a weak but interesting alignment between the ICD of I329L and the Toll protein of *Drosophila melanogaster*, as well as a plausible structural similarity between the ICD of I329L and the TIR domain of TLR3. Remarkably, the bioinformatics analysis presented here also demonstrates a region within the pI329L ICD similar to the BB loop of TLR3, the latter possibly functioning for interaction with TRIF.

In conclusion, the bioinformatics analysis suggested that (1) Both ECD and ICD domains of I329L could contribute to I329L-mediated inhibition of TLR3 signalling, (2) Signalling inhibition mediated solely by the pI329L ECD could result from interference with dsRNA binding via the DID and/or TLR homodimer formation via the PDD, and (3) Signalling inhibition mediated solely by the pI329L ICD could result from an interaction with the intracellular downstream signalling intermediate TRIF.

### 3.3. The I329L Extracellular Domain Interferes with TLR3 Signalling through a Putative Dimerization Domain (PDD)

To confirm whether the I329L PDD was responsible for inhibiting TLR3 signalling, we generated truncated I329L plasmids coding for either the entire extracellular domain plus the transmembrane region (ECDTM) or only the putative dimerization domain with the transmembrane region (PDDTM). These constructs encoded for stable expressed proteins with the expected size (Figure 3A). The function of both ECDTM and PDDTM was tested by IFN-β luciferase reporter assays in cells stimulated with the TLR3 ligand, Poly(I:C), and compared to the inhibitory potential of the full length I329L (Figure 3B). As expected, transfection of I329L strongly inhibited Poly(I:C)–induced activation of the IFN-β promoter when compared to cells transfected with the empty vector. The same suppressive effect was observed in cells transfected with either the ECDTM or the PDDTM regions of I329L.

When compared to control conditions, the expression of full length, ECDTM and PDDTM I329L constructs in cells stimulated with Poly(I:C) also clearly reduced the level of secreted IFN-β (Figure 3C), confirming the previous observation. By interfering with the TLR3 signalling pathway, preventing expression and secretion of type I IFN, the ASFV I329L protein successfully reduces the effectiveness of the host cell antiviral response. Our results thus indicate that the putative dimerization region of the I329L extracellular domain plays a critical role in this modulation.

### 3.4. I329L Functions as a General Inhibitor of TLR Signalling via Its PDDTM

An interesting question is whether I329L inhibition of TLR signalling is limited to TLR3 or, perhaps, might extend to other members of the TLR family of receptors. Interestingly, a number of amino-acid residues from the TLR3 dimerization domain are located in a conserved loop that can be observed in other TLR structures [15]. Superimposition of the I329L PDDTM in the TLRs’ structural models, as achieved above for TLR3, is shown for TLRs 4 and 5 (PDB IDs: TLR4—3FXI; TLR5—3J0A). Interaction between the PDDTM and TLR4 or TLR5 (or, indeed, possibly any TLR with which it interacts) would potentially abrogate the formation of TLR homodimers, preventing the initiation of their respective signalling mechanisms (Figure 4A,B).

As the I329L PDDTM region is capable of inhibiting dsRNA-mediated activation of the TLR3 pathway alone, we investigated its impact on other TLR pathways. Thus, the I329L PDDTM was tested for inhibition of TLR4, TLR5, TLR8 and TLR9 using an NF-κB responsive luciferase assay (PRDII-luciferase reporter). The cells were either stimulated with the corresponding ligand for each TLR or left untreated (details in Material and Methods). Comparison between samples expressing empty control vector only, full length I329L or I329L PDDTM showed a clear inhibition of all four TLRs tested (TLR4, TLR5, TLR8 and TLR9) (Figure 5A–D). Since the PDDTM construct consists of only a portion of the ECDTM, it necessarily acts upstream of TRIF and at the level of TLR ligand binding and/or dimerization.

### 3.5. Association of I329L and TLR3

The observation that the PDD of I329L inhibited signalling mediated by TLR3, 4, 5, 8 and 9 suggested the possible inhibition of TLR3 signalling through formation of a non-signalling TLR3-I329L heterodimer as the responsible mechanism. Our first attempts to demonstrate a TLR3-I329L association by immunoprecipitation using a lysis buffer containing Triton X-100 as the detergent were negative. Under these lysis conditions, only relatively strong protein–protein associations are detected. Digitonin, however, permits the detection of weak protein–protein interactions [17], and when the experiments were repeated replacing Triton X-100 with digitonin, a weak, but consistently detectable I329L was detected in the anti-TLR3 coprecipitate (the presence of TLR3 in the anti-I329L coprecipitate was weaker but positive), presumably reflecting the lower sensitivity of the TLR3 compared to I329L (Figure 6).

### 3.6. The I329L Protein Inhibits TLR3 and TLR4 Signalling through Direct Interaction with TRIF

The bioinformatics studies summarized above also suggested a functional role for I329L ICD in inhibiting TLR3 signalling, possibly by sequestering its downstream adapter, TRIF [3,7]. We built an expression plasmid coding solely for the I329L ICD and tested its effect on IFN-β induction by Poly(I:C) and TRIF. This construct encodes a stable protein with the expected size of ~20 kDa (Figure 7A). Luciferase reporter assays using the IFN-β promoter indicate that the I329L ICD is capable of inhibiting both extracellular (Poly(I:C)) and intracellular (ectopic TRIF) activation of IFN-β promoter (Figure 7B,C).

Importantly, TRIF is an adaptor protein controlling stimulation of both TLR3 and TLR4. and only the full length I329L and the ICD inhibit ectopically expressed TRIF (Figure 7C) suggesting the possibility of an I329L-TRIF interaction mediated by the ICD, leading to an inhibition of IFN-β suppression [3].

This suspicion of an interaction between ICD and TRIF was confirmed by analysis of both HA-ICD-I329L and TRIF immunoprecipitates (Figure 8). Thus, we propose a mechanism where the ICD of I329L directly interacts with TRIF, inhibiting the downstream signalling pathways of both TLR3 and TLR4.

## 4. Discussion

In this work, we define the mechanism of I329L, a virus inhibitor of TLR responses. Remarkably, I329L inhibits TLR responses by two entirely different mechanisms, one through its extracellular domain (ECD) and the other through its intracellular domain (ICD). The ASFV I329L gene codes for a Type I membrane protein which is highly conserved in the pathogenic and non-pathogenic virus, and which consists of a putative signal peptide (aminoacids 1–17), an extracellular N terminal domain (ECD), (aminoacids 18–239), a cytoplasmic domain (ICD) (aminoacids 261–329), and a transmembrane region (aminoacids 240–260). Subsequent work confirmed the inhibition of secreted IFN by I329L [3], and bioinformatics modelling [9] suggested that (1) The ECD of I329L might serve as a dsRNA decoy or hinder TLR3 signalling through formation of a I329L-TLR3 heterodimer, and (2) The ICD might inhibit through an impact on the downstream signalling intermediate TRIF.

We now demonstrate that I329L is a general inhibitor of TLR activation, with appropriate cellular localization on the surface membrane and in the early endosome. Furthermore, we present evidence supporting the claim that the ECDTM and ICD regions inhibit activation of IFN-β promoter and NF-κB by entirely different mechanisms, effectively preventing the induction of type I IFN and pro-inflammatory cytokines. Specifically, the ECDTM region inhibits activation of TLR3, TLR4, TLR5, TLR8 and TLR9 at the receptor level, whereas the ICD, but not the ECDTM, interferes with the adaptor TRIF, blocking intracellular activation of TLR3 and TLR4 signalling pathways.

The bioinformatics analysis of I329L identified a short sequence marginally homologous to the TLR dimerization domain, named the putative dimerization domain (PDD). Importantly, we show that this domain not only inhibits the activation of IFN-β promoter via TLR3, but also the activation of NF-κB by TLR4, TLR5, TLR8 and TLR9. On the basis of this broad specificity, we therefore hypothesized that the ECDTM of I329L might generally inhibit TLR activation through forming a non-signalling association with the TLR molecule via interaction of the PDD. The demonstration of a weak TLR3-I329L association by immunoprecipitation (in digitonin but not in Triton X-100) was consistent with this hypothesis.

We also reveal a previously unknown association between the ICD of I329L and TRIF, suggesting a role for I329L as a sequester of TRIF in the cytoplasm, thus preventing its interaction with signalling TLR3 or TLR4 homodimers. This hypothesis is supported by our experimental results and consistent with the previously observed inhibition of both TLR3 and TLR4 signalling by I329L [3], and the fact that both TLR3 and TLR4 signal via TRIF.

One area left unexplored is the precise function of the I329L DID region. Its structural homology to the ligand-binding region of TLR3 suggests that I329L could function as a competitor for binding to RNA or dsRNA. Further studies are required to explore this possibility, which could extend our understanding of I329L function and perhaps add another mechanism by which this viral protein prevents activation of TLR responses.

From the point of view of vaccine construction, we have further characterized the mechanism of I329L, a novel TLR antagonist encoded by the African Swine Fever Virus (ASFV) [3,7,9], as a prelude to its practical exploitation, for example, for the development of a possible attenuated virus vaccine for ASFV. Significantly, deletion of the I329L gene from an already attenuated ASFV strain, OURT88/3, led to a reduction on its capacity to induce protection against a highly virulent virus challenge [8]. This excessive attenuation with loss of capacity to induce protection has been described as a consequence of serial passage in cell cultures [18] or of successive gene deletion [19,20]. In addition, the deletion of I329L from a highly virulent strain, Georgia/2007, was not enough for attenuation [8]. Therefore, I329L is a player in the equilibrium between attenuation and protection and its characterization is certainly a valid contribution for the future development of an effective and safe vaccine against ASF.

The Toll-like receptor (TLR) family plays a crucial regulatory role in both the early innate and the later adaptive immune responses, driving pathogenic mechanisms in various immune-mediated and autoimmune diseases such as infection-associated sepsis, systemic lupus erythematosus, rheumatoid arthritis, and atherosclerosis. Manipulation of TLR responses offers an extraordinary opportunity for clinical intervention in health and disease [21,22,23,24,25]. For example, the development of structural analogues of TLR ligands or sequesters of adaptor proteins involved in TLR signalling is a considerable area of pharmaceutical research. Pathogens, such as ASFV, are known to develop strategies to suppress TLR signalling [26,27,28,29]. Our work clearly presents I329L as a viral immunomodulatory protein with at least two distinct mechanisms of action, and thus its potential as a therapeutic option for the control of excessive TLR activation provides an exciting avenue for future research.

## 5. Conclusions

In conclusion, we have characterized a novel activity of the ASFV I329L gene, a Type I membrane protein, which employs a versatile dual strategy for the inhibition of TLR responses. The extracellular domain (ECD) inhibits activation of TLR3, TLR4, TLR5, TLR8 and TLR9, through a short sequence homology to the conserved TLR dimerization domain, perhaps by mediating the formation of a non-signalling TLR3-I329L heterodimer. The intracellular domain (ICD) inhibits activation of TLR3 and TLR4 responses through its interaction with the intracellular signalling molecule TRIF. These properties not only reduce the effectiveness of the host innate immune response, but also considerably expand the potential of I329L as a ready-made tool for the manipulation of TLR responses.

## Figures and Tables

**Figure 1 viruses-15-00445-f001:**
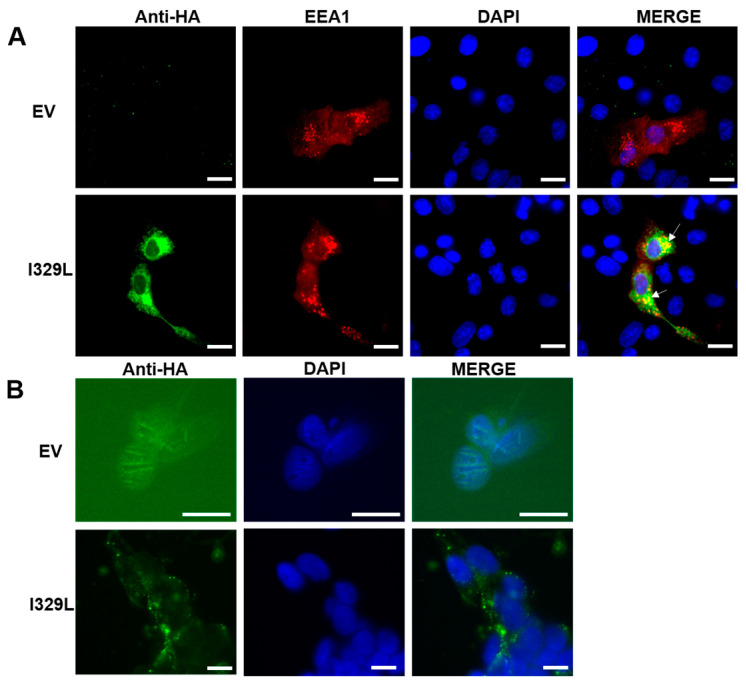
Localization of I329L to the early endosome and surface membrane. Indirect immunofluorescence was performed on: (**A**) Vero cells transfected with EEA1 Ct-tomato plasmid (to visualize early endosomes, shown in red) and pcDNA3-I329L-HA or empty control plasmid (EV), using an anti-HA antibody and anti-mouse cy5 conjugated secondary antibody to detect I329L expression (in green) and DAPI staining to define nuclei (in blue); and (**B**) In an alternative protocol, the cells were transfected with pcDNA3-I329L-HA or empty control plasmid (EV), and directly stained without permeabilization with an anti-HA antibody and anti-mouse Cy5 conjugated secondary antibody to detect I329L expression (in green) and counterstained with DAPI to define nuclei (in blue); Images were acquired with a 63× objective. Scale bars: 20 μm.

**Figure 2 viruses-15-00445-f002:**
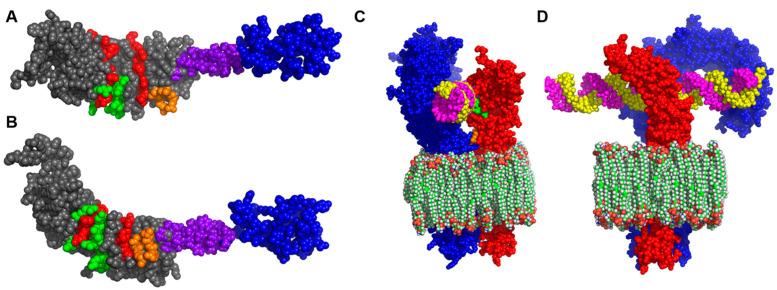
Bioinformatics modelling demonstrates shared functional homology of the extracellular domain of I329L with the ECD of TLR3. Models of I329L in a front (**A**) and side (**B**) view illustrate the structural features of the protein. The ectodomain is colored grey, the transmembrane region purple and the endodomain blue. The leucine rich repeats (LRRs) of the ectodomain are colored red. The aminoacids corresponding to the dsRNA interaction motif and the dimerization motif are colored green and orange, respectively. The I329L model was overlaid on one of the chains of the TLR3 dimer model (PDB: 3CIY) (to visualize the possible regions of interaction to form a heterodimer), shown in front (**C**) and side (**D**) view. In the dimer models, I329L is colored red, TLR3 blue and dsRNA yellow and pink.

**Figure 3 viruses-15-00445-f003:**
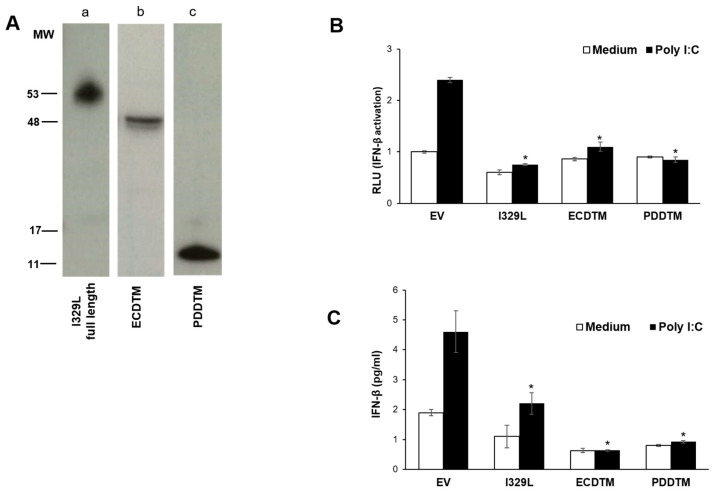
The ECD of I329L inhibits induction and expression of type I IFN. (**A**) HEK-293T cells were transfected with different HA-tagged I329L constructs and expression was confirmed by Western blot using an anti-HA HRP-conjugated antibody: (a) full length I329L, (b) the ectodomain with the transmembrane region (ECDTM), and (c) the putative dimerization domain with the transmembrane region (PDDTM). (**B**) HEK-293T cells were co-transfected with either pcDNA3HA (EV), pcDNA3-I329L-HA, or pcDNA3-ECDTM-HA, the IFN-β luciferase reporter and the β-galactosidase internal control plasmid. Cells were either induced with 25 μg/mL Poly(I:C) (TLR3 ligand) or left untreated (medium). Luciferase activity was normalized to β-galactosidase activity, and is shown as relative luciferase units (RLU). (**C**) HEK-293T cells were co-transfected with either empty vector pcDNA3HA (EV), pcDNA3-I329L-HA or pcDNA3-ECDTM-HA. At 48 h post-transfection, cells were either stimulated with 25 μg/mL Poly(I:C) for 5 h, or left untreated (medium). Secreted IFN-β concentration was determined by ELISA. Luciferase activity was normalized to β-galactosidase activity. Data are expressed as means of relative luciferase units (RLU) ± SD of triplicate culture wells and are representative of three independent experiments. RLU and IFN-β secretion ratios quantifying changes between Poly (I:C) treatment vs. untreated conditions for each of the constructs were compared against EV ratios and assessed by Student’s *t* test. * *p* ≤ 0.05.

**Figure 4 viruses-15-00445-f004:**
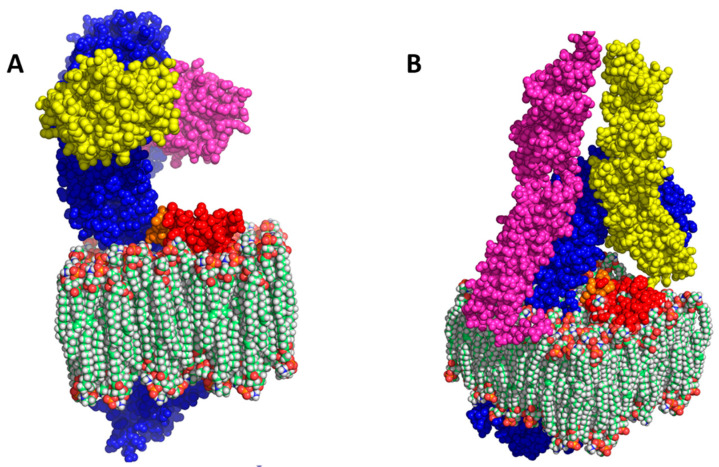
Bioinformatics modelling of I329L putative dimerization domain (PDDTM). A model of the PDDTM was created from the original I329L model and overlaid on one of the TLR dimer chains of TLR4 (**A**) (PDB: 3FXI) and TLR5 (**B**) (PDB: 3J0A). The structural similarity suggests an I329L-TLR interaction that would prevent the formation of signalling TLR homodimers. The I329L PDDTM is in red, with the dimerization motif in orange. TLR4 and TLR5 are in blue. The interacting molecules in the TLR4 model, visualized in yellow and pink, are the co-interactor MD2. In the TLR5 model, the same colors label flagellin (TLR5 ligand).

**Figure 5 viruses-15-00445-f005:**
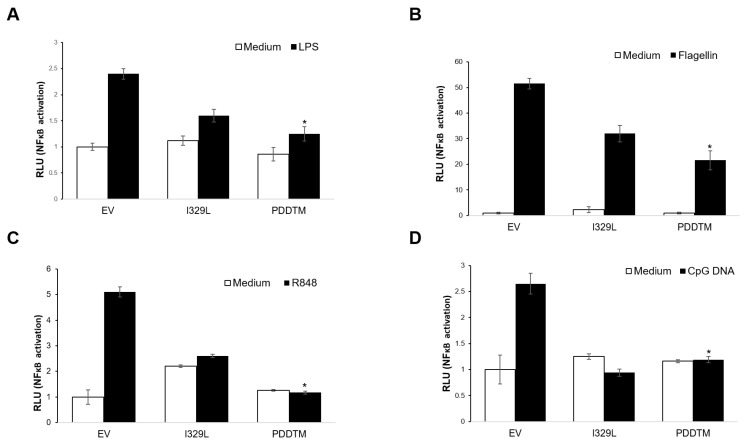
I329L is a general inhibitor of TLR signalling. (**A**) Vero cells were co-transfected with either the empty vector pcDNA3HA (EV), pcDNA3-I329L-HA, or pcDNA3-PDDTM-HA, the NF-κB luciferase reporter and the β-galactosidase internal control plasmid. Cells were induced with 100 ng/mL LPS (TLR4 ligand) or left untreated (medium). (**B**,**C**) HEK-293T cells were co-transfected with pcDNA3HA (EV) or pcDNA3-I329L-HA, or pcDNA3-PDDTM-HA, the NF-κB luciferase reporter, the β-galactosidase internal control plasmid, and TLR5, or TLR8 expression plasmids. Cells were either induced with 100 ng/mL Flagellin (TLR5 ligand) (**B**), 1 μg/mL R848 (TLR8 ligand) (**C**) or left untreated (medium). (**D**) Vero cells were co-transfected with either pcDNA3-HA (EV), pcDNA3-I329L-HA, or pcDNA3-PDDTM-HA, the NF-κB luciferase reporter and the β-galactosidase internal control plasmid. Cells were either induced with 10 μg/mL ODN2006 (TLR9 ligand) or left untreated (medium). Luciferase activity was normalized to β-galactosidase activity. Data are expressed as means of relative luciferase units (RLU) ± SD of triplicate culture wells and are representative of three independent experiments. RLU ratios quantifying changes between activation vs. untreated conditions for each of the constructs were compared against EV ratios and assessed by Student’s *t* test. * *p* ≤ 0.05.

**Figure 6 viruses-15-00445-f006:**
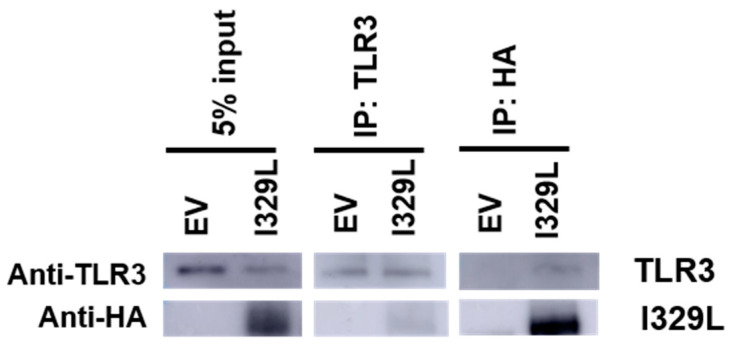
The I329L interacts with TLR3. HEK-293T cells stably expressing control and I329L were transfected with TLR3. After 48 h, cell lysates were subjected to immunoprecipitation using antibodies for TLR3 or HA (to capture I329L). The immunoprecipitates were then analyzed for the presence of I329L and TLR3 by Western blot.

**Figure 7 viruses-15-00445-f007:**
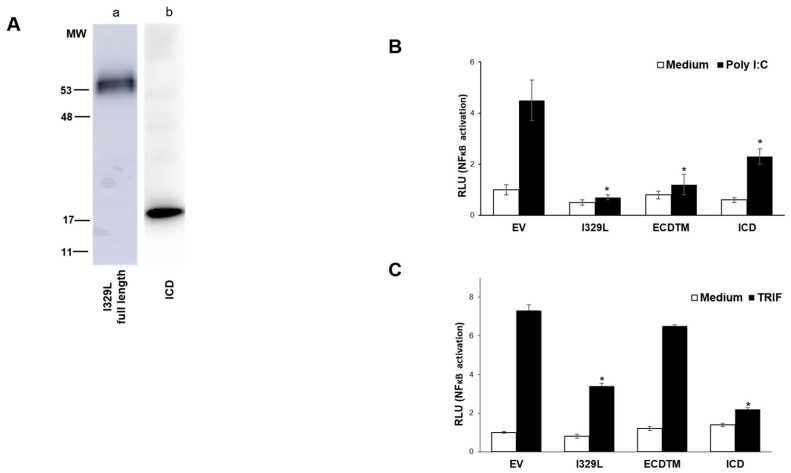
The ICD of I329L inhibits induction of type I IFN at the level of TRIF. (**A**) HEK-293T cells were transfected with different HA-tagged I329L constructs and expression was confirmed by Western blot using an anti-HA HRP-conjugated antibody: (a) full length I329L, (b) the intracellular domain (ICD). (**B**) HEK-293T cells were co-transfected with either pcDNA3HA (EV), pcDNA3-I329L-HA, or pcDNA3-ICD-HA, the IFN-β luciferase reporter and the β-galactosidase internal control plasmid. Cells were either stimulated with 25 μg/mL Poly(I:C) (TLR3 ligand) or left untreated (medium). (**C**) Luciferase activity was determined in HEK-293T cells co-transfected with pcDNA3HA (EV) or pcDNA3-I329L-HA, or pcDNA3-ECDTM-HA, or pcDNA3-ICD-HA, the IFN-β luciferase reporter and the β-galactosidase internal control plasmid. Cells were either induced with ectopically expressed TRIF or left untreated (medium). Luciferase activity was normalized to β-galactosidase activity. Data are expressed as means of relative luciferase units (RLU) ± SD of triplicate culture wells and are representative of three independent experiments. RLU ratios quantifying changes between activation vs. untreated conditions for each of the constructs were compared against EV ratios and assessed by Student’s *t* test. * *p* ≤ 0.05.

**Figure 8 viruses-15-00445-f008:**
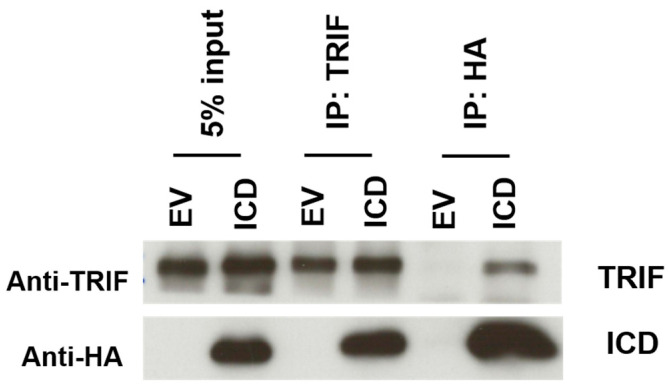
The ICD of I329L interacts with TRIF. HEK-293T cells were transfected with pcDNA3-HA (EV) or pcDNA3-ICD-HA (ICD) and pCMV-TRIF. After 48 h, cell lysates were subjected to immunoprecipitation using antibodies for TRIF or HA (to capture I329L ICD). The immunoprecipitates were then analyzed for the presence of I329L and TRIF by Western blot.

## Data Availability

No new data were created or analyzed in this study. Data sharing is not applicable to this article.

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
