# Peer review of "I329L: A Dual Action Viral Antagonist of TLR Activation Encoded by the African Swine Fever Virus (ASFV)"

_viruses, 2023, doi:10.3390/v15020445_

Round 1

Reviewer 1 Report

The manuscript describes a functional analysis of the mechanism by which the ASFV I329L protein, a putative TLR agonist,  inhibits induction of type I interferon. The results extend earlier studies by the same group on this protein by establishing that both the extracellular and intracellular domains, when coupled to the transmembrane domain can inhibit induction and secretion of IFN beta. A detailed predicted structure model of I329L alone or bound to TLR3 or dsRNA indicated potential modes of action of I329L.   The results showed that the intact protein or intracellular domain coupled to the transmembrane region could was sufficient to inhibit activation of several TLR proteins, TLR4, TLR5, TLR8 and TLR9 in addition to TLR3. Evidence for an inhibition of the interaction of TRIF with the cytoplasmic domain was provided by luciferase assays and by co-immunoprecipitation of TRIF with I329L cytoplasmic domain or intact protein but not the extracellular domain. Evidence for a weak interaction of I329L extracellular domain with TLR3 was obtained.

The paper is clearly written and the data support the conclusions drawn. The results provide novel data of interest in understanding how the I329L functions and further information of the diverse mechanisms by which ASFV inhibits the type I IFN response.

Specific points:

The figures don’t make clear what statistical comparisons are shown. This should be described in the Figure legends and text. For example in panel 3C it appears that the intact I329L protein inhibits secretion of type I IFN beta less well than either the ECDTM or PDDTM domains. Is this a reproducible and statistically significant result and if so is there a possible explanation?

Author Response

Comments and Suggestions for Authors

The paper is clearly written and the data support the conclusions drawn. The results provide novel data of interest in understanding how the I329L functions and further information of the diverse mechanisms by which ASFV inhibits the type I IFN response.

Response: We are heartened by the reviewer’s enthusiasm towards these data, and thank them for their careful review.

Specific points:

The figures don’t make clear what statistical comparisons are shown. This should be described in the Figure legends and text. For example in panel 3C it appears that the intact I329L protein inhibits secretion of type I IFN beta less well than either the ECDTM or PDDTM domains. Is this a reproducible and statistically significant result and if so is there a possible explanation?

Response: We have restructured parts of the figure legends to clarify that comparisons were made comparing the transfected constructs versus the ratios of activation/no treatment in control conditions (EV) in all cases, specifically: Fig 3, Fig 5, Fig 7. Regarding the apparent differences in inhibitory activity between the full-length I329L and the ECDTM/PDDTM constructs, no consistent increase in inhibitory activity was found for the separate domains over multiple assays

Reviewer 2 Report

Dear authors:

This manuscript described, on the one hand, that the ASFV I329L extracellular structural domain ECD inhibited the TLR3, TLR4, TLR5, TLR8 and TLR9 signaling pathways through a short sequence homologous to the conserved TLR dimerization domain (PDD), and that the PDD may mediate the weak association of I329L with TLR3 and its heterodimer formation. On the other hand, the ASFV I329L intracellular structural domain ICD was described to affect TLR3 and TLR4 signaling pathways through association with TRIF. Overall, this manuscript was logical and read with great fluency.

In my personal opinion, although I329L showed an inhibitory effect on the IFN response, more data are needed to support this view and if it is possibly to construct an I329L or its structural-domains-deficient ASFV strain for further evaluation. Following are my personal suggestions.

1. Does Figure 1B demonstrate that I329L is aggregating at the cell membrane? Is there a cell membrane probe or marker vector that probes the co-localization of I329L with cell membrane?

2. Figure 6 showed only weak interactions, which can be further demonstrated using exogenous interactions as well as colocalization. In addition, the HA detection in the right panel should not show two bands.

3. In Figure 6, if it is possible to use also exogenous interactions and colocalization to prove the interaction of ICD of I329L and TRIF?

4. Suggestions for the writing of this manuscripts: Is space needed in front of the unit? Such as Line 326 “25μg/ml Poly(I:C)”, line 370 “10μg/ml ODN2006 (TLR9 ligand)” and so on. Line 368 “pcDNA3HA (EV)”, is it right? There may be other writing problems in the manuscript that need to be improved and hope the authors can proofread it.

5. Did the title "3.6"of the manuscript is outlined appropriately?

Author Response

Comments and Suggestions for Authors

Dear authors:

In my personal opinion, although I329L showed an inhibitory effect on the IFN response, more data are needed to support this view and if it is possibly to construct an I329L or its structural-domains-deficient ASFV strain for further evaluation. Following are my personal suggestions.

Response: We thank the reviewer for their careful review and for the provided suggestions, and we have reworked several sections of the manuscript for increased clarity.

  1. Does Figure 1B demonstrate that I329L is aggregating at the cell membrane? Is there a cell membrane probe or marker vector that probes the co-localization of I329L with cell membrane?

Response: We were unable to pursue cell membrane co-staining when these experiments were performed; however, we hypothesize this is the case due to I329L’s detection in 1B without previous fixation or permeabilization. We have changed lines 233-235 to account for this.

  1. Figure 6 showed only weak interactions, which can be further demonstrated using exogenous interactions as well as colocalization. In addition, the HA detection in the right panel should not show two bands.

Response: Thank you for calling our attention to the mistake on figure 6. When the figure was prepared, the right panel introduced was not the correct one. Therefore, the figure was prepared again with the correct panel, which only has a band for HA.

  1. In Figure 6, if it is possible to use also exogenous interactions and colocalization to prove the interaction of ICD of I329L and TRIF?

 Response: We found that the I329L ICD construct has a broad cytoplasmic localisation due to the lack of membrane/targeting domains. Therefore, demonstrating its colocalization via immunofluorescence would be very challenging, and was not pursued due to the clear functional role demonstrated in luciferase assays.

  1. Suggestions for the writing of this manuscripts: Is space needed in front of the unit? Such as Line 326 “25μg/ml Poly(I:C)”, line 370 “10μg/ml ODN2006 (TLR9 ligand)” and so on. Line 368 “pcDNA3HA (EV)”, is it right? There may be other writing problems in the manuscript that need to be improved and hope the authors can proofread it.

Response: We have fully proofread the article and incorporated the suggested changes.

  1. Did the title"3.6"of the manuscript is outlined appropriately?

Response: We have clarified the title of section 3.6 to reflect the presented data.

Round 2

Reviewer 2 Report

Figure 6 still showed only weak interaction. The HA band in the left channel and the middle channel and the TLR3 band in the right channel were poorly specific and not clear. Can you improve it?

Author Response

The reviewer's observation is correct. The co-precipitation result for an I329L association relies on the detection of weak bands in the co-precipitation, as shown in Figure 6. This result, however, is highly significant as absolutely no evidence for an I329L-TLR3 interaction is observed with Triton X-100 lysates. This I329L-TLR3 interaction was only observed with a digitonin lysate, and the significance of this is clearly explained in the text (section 3.5) and the discussion (end of paragraph 3). Thus and according to the properties of these 2 detergents (Triton X-100 with digitonin) we may conclude that the I239L-TLR3 interaction is weak, as indeed we have observed.

We have now clarified this point by also emphasizing this in the conclusion of the revised version now submitted, and we hope that this now permits the acceptance of our work.